# Structural Variants of Dermatan Sulfate Can Affect the Expression of Proteins Involved in Breast Cancer Cell Survival

**DOI:** 10.3390/cells14201581

**Published:** 2025-10-11

**Authors:** Grzegorz Wisowski, Monika Paul-Samojedny, Katarzyna Komosińska-Vassev, Adam Pudełko, Ewa M. Koźma

**Affiliations:** 1Department of Clinical Chemistry and Laboratory Diagnostics, Faculty of Pharmaceutical Sciences in Sosnowiec, Medical University of Silesia in Katowice, Jedności 8, 41-200 Sosnowiec, Poland; kvassev@sum.edu.pl (K.K.-V.); mkozma@sum.edu.pl (E.M.K.); 2Department of Medical Genetics, Faculty of Pharmaceutical Sciences in Sosnowiec, Medical University of Silesia in Katowice, Jedności 8, 41-200 Sosnowiec, Poland; mpaul@sum.edu.pl; 3Genetic Laboratory Gyncentrum, ul. Wojska Polskiego 8a, 41-208 Sosnowiec, Poland; adampudelko@yahoo.pl

**Keywords:** cFLIP, dermatan sulfate, heme oxygenase-1, luminal breast cancer

## Abstract

Dermatan sulfate (DS) is an animal glycosaminoglycan with significant structural heterogeneity and a high, but variable density of negative electric charge. Owing to these characteristics DS displays a high degree of biological reactivity that is subject to regulation. We previously demonstrated that structural variants of DS rapidly induce moderate necroptosis in luminal breast cancer cells. In the present study, we investigated the intracellular molecular mechanism(s) that may underlie this effect, focusing on the expression of key regulators of intrinsic (BCL-2A1) and extrinsic (cFLIP) apoptosis, autophagy (Beclin-1), and oxidative stress protection (heme oxygenase-1 (HO-1)). Using RT-qPCR, Western blotting, immunofluorescence, and pharmacological inhibition, we have shown for the first time that DS, depending on its structure and the cancer cell line, can rapidly, albeit transiently, upregulate either the long or short cFLIP splicing variant and also reduce the level of HO-1. These effects are mediated via DS-triggered PI3K and/or NFκB signaling. Moreover, DS can also influence the intracellular distribution of these proteins. In contrast, this glycan did not affect the expression of *BCL-2A1* and *BECN1*. These findings indicate that DS induces coordinated molecular remodeling in luminal breast cancer cells that creates an intracellular environment favorable for necroptosis induction.

## 1. Introduction

Dermatan sulfate (DS) is an animal glycosaminoglycan (GAG) whose linear molecules (chains) are composed of repeating disaccharide units, each containing an N-acetylgalactosamine (GalNAc) residue and an iduronic acid (IdoA) or glucuronic acid (GlcA) residue. This GAG exhibits remarkable structural heterogeneity primarily due to variable and most likely tissue-specific patterns of disaccharide unit modifications including sulfation of unit components (mainly GalNAc) and C5 epimerization of precursor GlcA residues into IdoA residues (for details see [1,2,3]). The structural features of DS underlie significant biological reactivity, regulating the profile of this glycan interactions with multiple protein ligands [4], and involving it in the modulation of biological processes that remain incompletely understood. The importance of DS, highlighted in numerous investigations (see [5,6]), is further supported by its widespread distribution, primarily in the extracellular matrix of tissues, including the cancer microenvironment, where this glycan undergoes significant structural remodeling [7,8]. In tissues, DS occurs predominantly in the form of proteoglycans (PGs), in which one or more DS chains are covalently attached to various core proteins. However, as a result of extracellular processing of PGs, especially pronounced in cancer tissues [9], a significant pool of free DS chains can be released and become accessible to tumor cells, thereby affecting their behavior. Certain structural variants of non-neoplastic DS, including an isoform from human fibrosis-affected fascia (DF), which shares some structural features with the cancer-associated glycan, could quickly induce moderate necroptosis in the luminal breast cancer cell lines BT-474 and T-47D [10,11]. The underlying mechanisms involved at least two DS-triggered processes, including the activation of nuclear factor (NF)κB pathway, which exerted an inhibitory effect on the activation of necroptotic effector MLKL (mixed lineage kinase domain-like pseudokinase), and Rac1-mediated oxidative stress, which stimulated this lytic cell death [11]. However, the cellular context of the abovementioned events, which is linked to various processes that affect cancer cell viability and may be modulated by the examined DS variants, thus favoring the necroptosis induction, remains unknown. Such processes include apoptosis, which is a regulated cell death, and autophagy, representing a catabolic mechanism that enables the recycling of abnormal molecules and defected organelles to support cellular survival [12,13]. Moreover, cells have specialized defense mechanisms that allow them to prevent excessive oxidative stress and/or mitigate its consequences [14]. Therefore, we aimed to explore whether the structural variants of DS that previously effectively induced necroptosis could also influence the abovementioned processes. We addressed this objective by assessing in the DS-exposed luminal breast cancer cells the expression of key regulators of these processes, including B-cell lymphoma-2 (BCL-2)-related protein A1(BCL-2A1), cellular FLICE-inhibitory protein (cFLIP), Beclin1, and heme oxygenase-1 (HO-1). BCL-2A1 belongs to key inhibitors of the intrinsic apoptosis, acting by preventing the release of cytochrome c from mitochondria [15]. The protein is overexpressed in many tumors, including breast cancer, and this phenomenon has been associated with the development of tumor resistance to chemotherapy [15]. The influence of DS (or its structurally related counterpart, chondroitin sulfate (CS)) on the BCL-2A1 expression remains unclear. Nevertheless, it has been reported that, in colon cancer cells, CS can modulate the level of Bcl-2, a prototypical anti-apoptotic member of this family, in a manner probably dependent on this glycan structure [16,17]. In turn, cFLIP is a regulator of procaspase-8 activation/activity, thereby controlling the extrinsic pathway of apoptosis induction [18,19]. In cancers, cFLIP is recognized as an anti-apoptotic factor responsible for therapy resistance [19,20]. Interestingly, the downregulation of cFLIP significantly increased the sensitivity of luminal breast cancer cells MCF-7 to combined treatment with Smac and TNF-related apoptosis-inducing ligand, resulting in reduced cell viability and the enhanced activation of caspases-7/8 in them [21]. cFLIP, which is a product of *CFLAR* gene, occurs in three splicing isoforms, with the long cFLIP (cFLIP(L)) and short cFLIP (cFLIP(S)) variants being the most prevalent in human tissues [18,19]. The full-length cFLIP(L) consists of two death effector domains (DEDs) and a C-terminal caspase-like domain, whereas cFLIP(S) lacks this C-terminal fragment [18,19]. The impact of DS (or CS) on this protein is completely unknown.

Another protein selected for the investigation is Beclin 1, which is responsible for the formation of autophagosomes, i.e., double-membraned vesicles that engulf the cellular cargo destined for degradation during autophagy [13]. It has been proven that, upon oncogenic activation, autophagy can sustain and promote cancer development by providing nutrients, released during self-digestion, as well as by eliminating defected organelles, especially mitochondria [13]. An impact of DS on the Beclin-1 metabolism in cancer cells has not been tested to date. However, it was shown that the DS-bearing proteoglycan, known as decorin, was able to enhance Beclin-1 expression, and this effect was mediated via its interaction with VEGFR2, with subsequent upregulation of paternally expressed gene-3 in endothelial cells [22]. In addition, lysosomal overloading with DS in an animal model of mucopolysaccharidosis VI led to impaired autophagy and an increased level of Beclin-1 in the affected organs [23].

The last of the chosen proteins is HO-1, also known as heat shock protein 32, which is a significant element in the cellular stress response [24]. This enzyme, which catalyzes the first step of heme catabolism, via products of this reaction, regulates many processes related to the cell behavior such as proliferation, survival, or migration [25]. Thus, it is believed that HO-1 may be involved in tumor initiation, progression, or invasion [25], although the role of this protein in the pathogenesis of breast cancer seems to be more complex and ambiguous [26]. However, it has recently been reported that pharmacological inhibition of HO-1 in MCF-7 luminal breast cancer cells can either reduce cell viability [27] or induce apoptosis through oxidative stress [28], depending on the inhibitor used. To date, the impact of DS on the HO-1 expression in cancer cells has not been investigated. However, it has been shown that CS can upregulate the enzyme during the H_2_O_2_-induced oxidative stress in human neuroblastoma cultures [29].

## 2. Materials and Methods

### 2.1. The Used Structural Isoforms of DS

In this investigation, we used three structural variants of dermatan sulfate that were previously characterized in terms of their structural features [10]. These isoforms included the glycan from porcine intestinal mucosa (PM, commercial preparation Cat#C4384, Sigma-Aldrich, Burlington, MA, USA) and two variants of human origin, i.e., DS from fibrotic fascia (DF) and from normal fascia (NF). The protocol of this study was approved by the local Bioethics Committee of the Medical University of Silesia in Katowice (permission no.: PCN/CBN/0052/KB1/52/22).

### 2.2. Cell Cultures

Both luminal breast cancer cell lines BT-474 and T47D were obtained from the American Type Culture Collection (Manassas, VA, USA). The cells were grown, respectively, in DMEM/F12 (Cat# D8437, Sigma-Aldrich, St. Louis, MO, USA) or RPMI-1640 (Cat# R8758, Sigma-Aldrich, St. Louis, MO, USA) medium supplemented with 10% fetal bovine serum (FBS; Cat# S181H, Biowest, Riverside, MO, USA), MycoZap Plus-CL (Cat# VZA-2012, Lonza, Rockville, MD, USA), and insulin (5 μg/mL for BT-474 and 10 μg/mL for T47D) (Cat# BE02-033E20, Lonza, Rockville, MD, USA). The cells were allowed to grow at 37 °C in a 95% humidified atmosphere with 5% CO_2_.

### 2.3. Quantitative Analysis of Gene Expression on the RNA Level

The cancer cells were seeded into a 24-well plate at a density of 18,000 cells per well and grown for 24 h in the complete medium. Then, the growing medium was exchanged with that containing 0.5% FBS, and cells were left to grow for another 24 h. Subsequently, the cells were exposed for three hours to an individual DS variant at a concentration of 2.5 or 25 µg/mL. The isolation of total RNA using the NucleoZOL reagent (Macherey-Nagel, Allentown, PA, USA) was conducted as described previously [11]. The mRNA copy number for *BCL2A1*, *BECN1*, *CFLAR*, or *HMOX1* was determined using RT-qPCR based on the specific primers (KiCqStart^®^ SYBR^®^ Green Primers, Merck, Darmstadt, Germany) with sequences that are presented in Table 1, according to the manufacturer’s protocol.

The gene expression was examined as described previously [11].

### 2.4. Analysis of Protein Expression and Phosphoinositide 3-Kinase Activation by Western Blotting

The BT-474 or T-47D breast cancer cells were seeded into a six-well plate (Corning Incorporated, New York, NY, USA) and exposed to an individual DS variant at a concentration of 25 µg/mL for several incubation periods. After intense rinsing, the cells were suspended by scraping in RIPA buffer (40 mM TrisHCl, pH 7.5, 0.15 M NaCl, 0.002 M EDTA, 0.5% Igepal CA630 (Sigma-Aldrich, Saint Louis, MO, USA), 0.5% sodium deoxycholate (Sigma-Aldrich), and 0.1% sodium dodecyl sulfate (SDS, Sigma-Aldrich)), containing protease inhibitor cocktail (Merck, Darmstadt, Germany)). The extracted proteins were quantified using a Pierce BCA Protein Assay Kit (Thermo Fisher Scientific, Waltham, MA, USA) and subjected to SDS-PAGE, followed by Western blotting as described previously [11]. The used primary antibodies were as follows: the rabbit CFLAR/FLIP polyclonal antibodies (Cat# 10394-1-AP, Thermo Fisher Scientific) at a dilution of 1:1000; the rabbit HO-1/HMOX1 polyclonal antibodies (Cat# 10701-1-AP, Thermo Fisher Scientific), diluted to 1:3000; the rabbit phospho-phosphoinositide 3-kinase (PI3K) p85 (Ty458) polyclonal antibodies (Cat. #4228, Cell Signaling Technology, Danvers, MA, USA) at a dilution of 1:1000; or the rabbit polyclonal GAPDH antibodies (#2275, Trevigen, Gaithersburg, MD, USA), diluted to 1:2500. The horseradish peroxidase-conjugated goat anti-rabbit antibodies (#A9169, Merck) at a dilution of 1:12,000 were used as secondary antibodies. cFLIP was detected by chemiluminescent reaction using the Pierce ECL Western Blotting Substrate (Thermo Fisher Scientific) followed by detection with Alliance Q9 Advanced imaging system (Uvitec, Cambridge, UK). The remaining proteins were detected using a colorimetric 3,3’,5,5’-tetramethylbenzidine substrate (Thermo Fisher Scientific).

### 2.5. Analysis of Protein Level in Breast Cancer Cells by Immunofluorescence

The BT-474 or T47D cancer cells were seeded into eight-well glass chamber slides (Cat# 154534, Thermo Fischer Scientific) at a density of 6500 cells per well and cultured for 24 h in the complete medium. Then, a medium was replaced with a medium containing 0.5% FBS, and the cells were allowed to grow for another 24 h. After this incubation, the cells were exposed to the tested DS variants at a concentration of 25 µg/mL for two or three hours. In some experiments, the DS variants were used in combination with 100 nM wortmannin (Cat# 1232, Tocris, Bio-Techne, Bristol, UK) or 40 µM cardamonin (Cat# 2509, Tocris). Subsequently, the cells were fixed and permeabilized as described previously [11]. The cells were then incubated overnight with the rabbit CFLAR/FLIP polyclonal antibodies (#10394-1-AP, Thermo Fisher Scientific) at a dilution of 1:100 or the rabbit HO-1/HMOX1 polyclonal antibodies (#10701-1-AP, Thermo Fisher Scientific), diluted to 1:200. The Alexa Fluor Plus 555-conjugated goat polyclonal anti-rabbit IgGs (Cat# A32732, Thermo Fisher Scientific) at a concentration of 3 µg/mL were used as a secondary antibodies. Finally, the cells were stained with Hoechst 33342 (Cat# H3570, Thermo Fisher Scientific), and their images were captured by a Leica DMI 6000B microscope (Leica Microsystems GmbH, Wetzlar, Germany). The images were quantified using Leica AS hardware version 3.2.1.9702. (GmbH, Wetzlar, Germany).

### 2.6. Statistical Analysis

The data were analyzed using the Statistica 13.3 application (TIBCO Software Inc., Cracow, Poland). The normality of the distribution was verified using the Shapiro–Wilk test, and the variance homogeneity was then analyzed using the Levene’s test. The data were presented as the mean ± standard deviation (SD) for the Western blotting analysis and the quantification of gene expression by the mRNA level or as the mean ± standard error of the mean (SEM) in the case of cell culture experiments. The between-group comparisons were assessed using a one-way ANOVA and the post hoc Tuckey’s test, with *p* ≤ 0.05 as being significant. For the non-parametric data, the between-group differences were estimated using the Kruskal–Wallis test by ranks, with *p* ≤ 0.05 as being significant.

## 3. Results

### 3.1. DS Affects the Expression of Some Genes Involved in Cell Survival

Our previous papers [10,11] have shown that some structural variants of DS, i.e., NF and PM or PM and DF, could quickly induce necroptosis of moderate intensity in the primary breast cancer cell line BT-474 or the metastatic breast cancer cell line T-47D, respectively. Moreover, at least in the T-47D cells, the underlying mechanisms also included a stimulatory effect of the tested DS variants on the expression of the gene for receptor-interacting protein kinase (RIPK) 3 [11], which is directly responsible for the activation of the necroptotic executor—MLKL—in the death cascade [12]. Thus, in the present study, we further explored molecular mechanisms that may facilitate the DS-dependent induction of necroptosis, assessing the impact of these biologically active structural variants of the glycan on the expression of genes, whose products are implicated in cell survival via the regulation of intrinsic (BCL-2A1) or extrinsic (cFLIP) pathways of apoptosis, autophagy (Beclin-1), and oxidative stress defense (HO-1). The expression of these genes was analyzed in the breast cancer cells that had been exposed to the variants for three hours because, after such an incubation period, the effects of these glycans on both the *RIPK3* upregulation and MLKL activation were observed in our previous studies [11]. Moreover, the tested DS variants were used at two concentrations, i.e., 25 µg/mL and 2.5 µg/mL, to assess the biological efficacy of each glycan in terms of its impact on gene expression. Interestingly, when these DS variants were previously tested at a higher concentration, they effectively triggered necroptosis of breast cancer cells [10,11]. The conducted investigations demonstrated that, in BT-474 cells, none of the tested DS variants at either concentration affected the level of mRNA for BCL-2A1, Beclin-1, and HO-1 (Figure 1A–D). Only PM, when it was applied at 25 µg/mL, was able to weakly (1.4 fold) yet statistically significantly stimulate the expression of *CFLAR* in these cells (Figure 1C). Similarly, in T-47D cells, neither PM nor DF influenced the expression of both *BCL-2A1* and *BECN1* (Figure 2A,B). However, both DS variants, when they were used at 25 µg/mL, upregulated the *CFLAR* expression by approximately 2.5 times in these cells (Figure 2C). In addition, unexpectedly at this concentration, both PM and DF were able to significantly increase the levels of *HMOX1* mRNA (Figure 2D).

### 3.2. DS Can Affect the Expression of Both CFLAR and HMOX-1 on the Protein Level

To examine whether DS could rapidly increase the expression of cFLIP and HO-1, we assessed by the Western blot analysis the content of the proteins in the lysates of BT-474 or T-47D cells that had been exposed for three or six hours exclusively to the DS variants, effectively upregulating the gene transcription (i.e., PM in BT-474 cells or DF and PM in T-47D cultures) (Figure 1 and Figure 2). We chose such incubation periods also considering the rapid DS-dependent induction of necroptosis, which was previously detected after three hours of the breast cancer cell exposure [10,11]. Therefore, if cFLIP and HO-1 contribute mechanistically to this DS-triggered cell death, their expression should be altered within a comparable time frame. As shown by the immunoblot results (Figure 3A,B and Figure 4A,B), the examined DS variants can affect cFLIP levels only in these luminal breast cancer cell cultures that had been exposed to them for three hours, but the effects were strongly dependent on both glycan structure and cell line. When compared to the untreated control, PM increased the amount only of cFLIP(L) as indicated by the molecular weight of the component, demonstrating the reactivity with anti-cFLIP antibody (Figure 3A,B and Figure 4A,B), but this impact was significant solely in the BT-474 line (Figure 3A,B and Figure 4A,B). In contrast, DF had no effect on the cFLIP(L) level but strongly upregulated the protein short form (cFLIP(S)) in the T-47D cultures (Figure 4A,B), which was manifested by the appearance of this isoform band only on the blots of DF-treated cells (Figure 4A). The significant stimulating effect of the abovementioned DS variants on the cFLIP content in the breast cancer cells after three hours of incubation was also confirmed by the immunofluorescence data (Figure 3C,D and Figure 4C,D), although the alterations in this parameter were less pronounced than those observed in the Western blotting analysis, especially in the case of the BT-474 cells (Figure 3B versus Figure 3D). However, in contrast to this last method, immunofluorescence revealed a significant increase in the cFLIP level also in the PM-treated T-47D cells (Figure 4B versus Figure 4D). The reason(s) for these discrepancies between Western blotting and immunofluorescence data remains unknown. Nevertheless, since the results obtained using both approaches were consistent for the remaining DS variants, we selected immunofluorescence for further examination on the detailed dynamic of the DS-dependent effect on cFLIP expression prior to the observed protein accumulation. This choice was dictated by the lower requirement of this method as to the amount of DS used, which is particularly relevant in the case of the human-derived DF variant. The results of this kinetic analysis (Figure 3D and Figure 4D) along with the Western blot data obtained for cells that were exposed to the tested glycans for six hours (Figure 3B and Figure 4B) support the conclusion that, regardless of the DS variant or cancer cell line used, this glycan induced only a sudden and transient upregulation of cFLIP, which appears after three hours of incubation.

Using immunofluorescence, we also assessed the level of nuclear translocation of cFLIP in the breast cancer cells that were exposed to the tested DS variants for three hours, ensuring a strong effect of these glycans on the protein expression, because subcellular distribution of this molecule is important for its biological functions [30,31]. According to the obtained data (Figure 3D and Figure 4D), none of the examined glycans exerted a significant influence on the nuclear localization of cFLIP, which supports the possibility of cytosolic accumulation of protein in the exposed cells.

Based on the results of immunoreactivity with the anti-HO-1 antibody of detergent-extracted species (Figure 5A,E and Figure 6A,E), it can be assumed that both the control and the DS-exposed breast cancer cells contain a significant amount of a component with a molecular weight of ~28 kDa, which is presumably a truncated form of HO-1, in addition to a much smaller level of the 32 kDa component, which likely corresponds to the full-length protein [24,32]. However, in contrast to the impact on cFLIP, none of the DS variants significantly influenced the level of HO-1 in either breast cancer cell lines after both three or six hours of incubation (Figure 5B and Figure 6B). These observations were further supported by the results of immunofluorescence analysis (Figure 5C and Figure 6C). Therefore, we decided to examine the content of HO-1 in the breast cancer cells that were exposed to the tested DS variants for a shorter time, i.e., one or two hours, using immunofluorescence. We found no alterations in the HO-1 level at either time point in the BT-474 cells exposure to PM (Figure 5C,D). In contrast, unexpectedly, DF, when present for one hour in the cultures of T-47D cells, slightly but significantly upregulated HO-1 (Figure 6C), which may be a mechanism for alleviating oxidative stress that was previously observed to be rapidly triggered by this variant [11]. In turn, after two hours of exposure, both PM and, especially, DF effectively reduced the HO-1 level in the T-47D cells compared to the untreated controls (Figure 6C,D).

Since the truncated HO-1 can be translocated to nuclei, where it modifies gene transcription [32], we investigated by immunofluorescence whether the examined DS variants could also affect the nuclear level of this protein in the breast cancer cells that were cultured for two hours of treatment. As evidenced by the data obtained (Figure 5C and Figure 6C), the observed effects were rather cell type- than DS type-dependent. This phenomenon was manifested by a significant decrease in the nuclear content of the protein only in the T-47D cells that were exposed to each of the tested glycans (Figure 5C versus Figure 6C).

### 3.3. PI3K and/or NFκB Signaling Pathways Mediate the DS-Induced Expression of cFLIP and HO-1

Both PI3K and NFκB play crucial roles in the biology of cancer cells regulating many aspects of their behavior including cell survival [33,34]. We have previously shown that all of the currently tested DS variants effectively stimulated the activation of NFκB, significantly increasing its nuclear translocation within one hour of incubation [11]. However, the influence of these glycans on PI3K phosphorylation and the dynamics of this process remain unknown. Western blotting analysis (Figure 7A,B and Figure 8A,B) clearly shows that all of the tested DS variants, when compared to the controls, could significantly modify the kinase activation, demonstrating a similar kinetic profile in both studied breast cancer cell lines. This phenomenon was manifested by a significant increase in the phospho-PI3K level after 20 min of exposure that was followed by a substantial reduction in the content of this form of kinase at 35 min (Figure 7B and Figure 8B). Thus, in order to examine the involvement of the PI3K or NFκB pathways in mediating the DS-induced expression of cFLIP in the breast cancer cell lines, we analyzed the protein levels in cultures that had been treated with an individual DS variant in a combination with a pharmacological inhibitor of the respective pathway using immunofluorescence. We used 100 nM wortmannin to block PI3K and 40 µM cardamonin to inhibit NFκB. In addition, the cFLIP levels were assessed after three hours of treatment, which was the optimal time for the DS-dependent stimulation of this protein expression to occur. The obtained data (Figure 7C and Figure 8C) elicited interesting differences in the regulation of cFLIP expression between the BT-474 and T-47D cells, as evidenced by the distinct effects of wortmannin and cardamonin on the protein level in the inhibitor-treated versus untreated controls of each of these lines (Figure 7C and Figure 8C). In BT-474 cells, wortmannin, when was used alone, significantly increased the cFLIP level compared to untreated cells, whereas no such effect was observed in the T-47D line (Figure 7C and Figure 8C). In contrast, cardamonin upregulated the protein only in the T-47D cells (Figure 7C and Figure 8C). Nevertheless, both wortmannin and cardamonin effectively suppressed the DS-induced increase in the cFLIP content in both cell lines (Figure 7C and Figure 8C), indicating the involvement of both PI3K and NFκB signaling pathways in mediating this cell response.

To explore the signaling mechanism that may be responsible for promoting the DS-dependent downregulation of HO-1 in T-47D cells, we focused only on the PI3K pathway. This decision was based on the very short time interval between the examined DS variant-induced downregulation of HO-1 (two hours of incubation) and the previously observed [11] maximum effect of the glycans on the nuclear translocation of NFκB (one hour of exposure) in these cells, which made a direct regulatory link between these processes unlikely. As results from the obtained data (Figure 8D), wortmannin completely blocked the impact of both DS variants on HO-1 expression in T-47D cells, suggesting that PI3K signaling is involved in regulating this process. In turn, wortmannin did not alter the neutral effect of PM on the HO-1 level in BT-474 cells, indicating that this variant does not exert any influence on the protein via the PI3K pathway in these cells (Figure 7D).

## 4. Discussion

To gain deeper insight into the mechanism(s) triggered by the DS variants, which were previously capable of inducing moderate necroptosis rapidly, i.e., after approximately four hours of incubation, in luminal breast cancer cells [10], we investigated the effects of these glycans on the expression of several proteins, implicated in the regulation of cell survival. We found that none of the DS variants tested in either of the studied cell lines affected the mRNA levels of *BCL-2A1* or *BECN1*. This observation suggests that, during a short-term exposure, the tested DS variants may not influence the expression of key regulators of the intrinsic pathway of apoptosis or autophagy in luminal breast cancer cells, and therefore do not directly modulate these processes per se. This possibility is at least partially supported by our previous findings, which demonstrated that these DS variants did not also alter the activity of effector caspases-3/7 during short exposure periods [10]. In contrast, with the exception of NF, the tested DS variants when used at a higher concentration (i.e., at 25 µg/mL) significantly upregulated *CFLAR* in both BT-474 and T-47D cells. Furthermore, in parallel with the increase in *CFLAR* mRNA levels, the tested variants also elevated the cellular abundance of cFLIP, although the results of Western blotting analysis suggest that the sensitivity of different breast cancer cell lines to the same DS structure may vary with respect to this last effect. Nevertheless, the observed concordance between the DS-induced impact on both the *CFLAR* mRNA and cFLIP content suggests that an accumulation of the protein in both cancer cell lines results primarily from the stimulation of its synthesis rather than impaired degradation. To the best of our knowledge, our study is the first to demonstrate a regulatory link between glycosaminoglycans, in particular DS, and cFLIP expression. Notably, although the DS variant-dependent increase in cFLIP level was only transient in luminal breast cancer cells, it temporally correlated with the dynamic of the DS-induced activation of the necroptotic effector MLKL as previously observed in these cells [11]. In the present study, we also demonstrated that the tested DS variants selectively promoted the expression of specific cFLIP splicing isoforms. This was reflected by the accumulation of the cFLIP(L) in all cell cultures that were treated with the PM variant and the selective upregulation of cFLIP(S) in the DF-treated T-47D line. The DS variants used in this study differed substantially in their structural features. The PM isoform contains up to six times fewer disaccharides with the GlcA residue than the DF isoform [10]. Moreover, although all tested variants are predominantly composed of 4-O-sulfated disaccharides (with DF containing fewer of those components than PM), they differ in the profiles of the remaining disaccharide types [10]. The PM variant is enriched in 6-O-sulfated and 2,6/4,6-O-disulfated units, whereas DF is almost completely devoid of disulfated disaccharides but is characterized by the highest proportion of unsulfated units, accompanied by 6-O-sulfated [10]. These structural distinctions likely underlie the observed isoform-specific effects, suggesting a strong regulatory relationship between DS structure and cFLIP mRNA processing in luminal breast cancer cells. In contrast to the impact on cFLIP, the DS-triggered effect on the HO-1 level was more complex and clearly dependent, at least in part, on the type of breast cancer cells. A significant response was detected only in the T-47D cultures, as reflected by the upregulation of *HMOX1* after three hours of exposure, which unexpectedly coincided with the unaltered level of this protein in the treated cells compared to the controls. However, following a shorter, two-hour incubation, both the PM and DF variants markedly reduced the content at least of the truncated isoform of HO-1 in the T-47D cells. We hypothesized that this phenomenon may reflect a direct effect of DS on HO-1, followed by a compensatory cellular response manifested as the upregulation of *HMOX1* expression, ultimately leading to the normalization of protein level. On the other hand, oxidative stress is a well-established stimulator of HO-1 expression [24]. However, although the DF variant was able to slightly increase the HO-1 level in the T-47D line after just one hour of incubation, at this time the PM isoform exerted no effect on the protein expression in either BT-474 and T-47D cells despite the presence of moderate oxidative stress, which had previously been shown to be induced by all the tested glycans during the first hour of exposure [11]. This stress response was found to be mediated via a mechanism requiring the Rac1 activity [11]. Taken together, these results suggest that not only the redox imbalance per se, but also its intensity and/or nature of triggering stimulus, i.e., the DS structure, may be a critical determinant of HO-1 expression. Moreover, our current results clearly indicate that the downregulation of HO-1 that was observed in T-47D cells after two hours of exposure to both PM and DF variant was mediated via the DS-dependent activation of PI3K. The kinase is also known to act downstream to the CS-triggered upregulation of HO-1 in neuroblastoma cells and the simvastatin-dependent stimulation of this protein expression in colon cancer cells [29,35]. However, we also showed that, although PM stimulated the activation of PI3K in BT-474 breast cancer cells, it induced any measurable alterations in the HO-1 content during the short-term exposure. All of these findings suggest that the involvement of the PI3K pathway in regulating the HO-1 expression is context-dependent, and may vary according to both the cell type and the specific nature of triggering factor. Nevertheless, both the PI3K and NFκB pathways were involved in DS variant-induced signaling events that modulated cFLIP expression in luminal breast cancer cells. The participation of the aforementioned pathways in regulating cFLIP expression has also been confirmed in various experimental models [36]. Our results derived from experiments using pharmacological inhibition indicate that both of these signaling pathways can drive both the PM-triggered expression of cFLIP(L) as well as the DF-induced synthesis of cFLIP(S) in luminal breast cancer cells. Interestingly, a previous study on highly aggressive glioblastoma multiform cells has demonstrated that the translation of cFLIP(S) is controlled via the Akt–mammalian target for rapamycin (mTOR)–p70S6 kinase 1 (S6K1) signaling axis [37]. Nevertheless, based on our current and previous results [11], it can be hypothesized that NFκB signaling contributes more predominantly than the PI3K pathway into the PM variant-induced upregulation of cFLIP(L). This hypothesis is supported by the observation that PM upregulated the protein more strongly in BT-474 cells than in T-47D cells, and this effect was paralleled with the more robust PM-dependent activation of NFκB relative to PI3K in BT-474 cells compared to T-47D [11].

Therefore, based on the results of the present study, the following important question arises: does the DS-dependent remodeling of expression profiles for cell survival-related proteins create an intracellular environment that facilitates the induction of necroptosis by specific glycan variants in luminal breast cancer cells? The canonical function of cFLIP is to modulate procaspase-8 activation, being a critical event in both the death receptor-induced apoptosis and death receptor-independent apoptosis via a platform called ripoptosome, the assembly of which is under the control of RIPK1 [19]. Depending on its splicing isoform, cFLIP can either (1) block the recruitment of procaspase-8 to the death complexes and completely inhibit its processing/activation (as in the case of cFLIP(S)) or (2) bind and activate procaspase-8 without it processing (as occurred with cFLIP(L)), which restricts proteolytic activity of the zymogene only to local substrates, such as neighboring procaspase-8 or RIPK1 molecules within the death complex [18,19,36]. As such, the ratio between cFLIP(L) and procaspase-8 in the death complexes is a decisive factor in inducing apoptosis, inhibiting necroptosis due to RIPK1 cleavage, or protecting RIPK1, which can then form along with RIPK3 a platform for the activation of the necroptotic executor MLKL [18,19,36]. Thus, we hypothesized that both the upregulation of cFLIP(S) by DF in T-47D cells and the significant accumulation of cFLIP(L), especially seen in the PM-treated BT-474 line, may promote conditions that suppress apoptosis while preserving RIPK1. This in turn may favor the induction of necroptosis. These pro-necroptotic conditions may be further reinforced in T-47D cells by the DS variant-triggered downregulation of HO-1. This hypothesis is supported by reports of a possible regulatory relationship between HO-1 and RIPK3, in which the downregulation of one protein co-occurs with the upregulation of the other, as observed in various cell models [38,39]. Additional mechanism(s) determining the fate of breast cancer cells exposed to DS variants may also result from the non-canonical functions of cFLIP and/or HO-1. These functions, which often require nuclear translocation of these proteins, include modulation of the activity and/or stability of key transcriptional factors such as NFκB (regulated by both cFLIP and truncated HO-1) [19,32,40] or nuclear factor erythroid 2-related factor-2 (Nrf2) (which is affected by HO-1) [24,32]. Moreover, cFLIP(L) has been implicated in the stimulation of Wnt signaling [30]. Thus, all of these non-canonical functions contribute in the promotion of pro-survival cellular programs. Unfortunately, due to financial limitations, we were unable to assess the subcellular distribution of cFLIP and HO-1 in the DS-exposed cancer cells by Western blot analysis. Nevertheless, our immunofluorescence data indicate that none of the applied DS variants altered the nuclear localization of cFLIP, and this observation is particularly relevant to cFLIP(L), which is known to undergo such a translocation [30]. Thus, this observation suggests that the PM variant, which significantly upregulated cFLIP(L), may also control its subcellular distribution by promoting cytosolic accumulation. This localization pattern of cFLIP(L) could be more favorable for the induction of necroptosis than nuclear localization as the cytosolic protein participates directly in the death complex assembly. On the other hand, our immunofluorescence analysis revealed that the levels of nuclear HO-1 were significantly reduced in the T-47D cells that were grown for two hours in the presence of PM or DF, compared to untreated controls. Notably, this decrease coincided with a similar reduction in HO-1 content in the cell lysates, suggesting that the observed nuclear depletion is more likely a consequence of the DS-induced downregulation of the protein rather than the direct inhibition of its nuclear translocation. However, the observed decreased level of nuclear HO-1 may facilitate the DS variant-driven necroptosis by impairing of the Nrf2-dependent transcriptional program in the exposed cells [24,32].

Our findings clearly indicate that the DS-induced necroptosis in luminal breast cancer cells results from the multifaceted impact of this glycan on various components of cellular machinery, which is only partially delineated by the present study. Therefore, further explorations are warranted to elucidate the detailed molecular mechanisms through which this naturally occurring glycan modulates cancer cell fate. Importantly, necroptosis represents a form of lytic cell death that promotes inflammation and may contribute to the breaking of the immunological tolerance of the host to an established tumor. As such, the ability of DS to induce necroptosis highlights its potential as a therapeutic agent particularly if the relationship between this glycan structure and biological properties becomes better understood. This challenge seems to be getting closer to being overcome, since the dynamic development of various analytical methods, allowing the examination of the DS sequence, has been observed in recent years [41,42,43].

## Figures and Tables

**Figure 1 cells-14-01581-f001:**
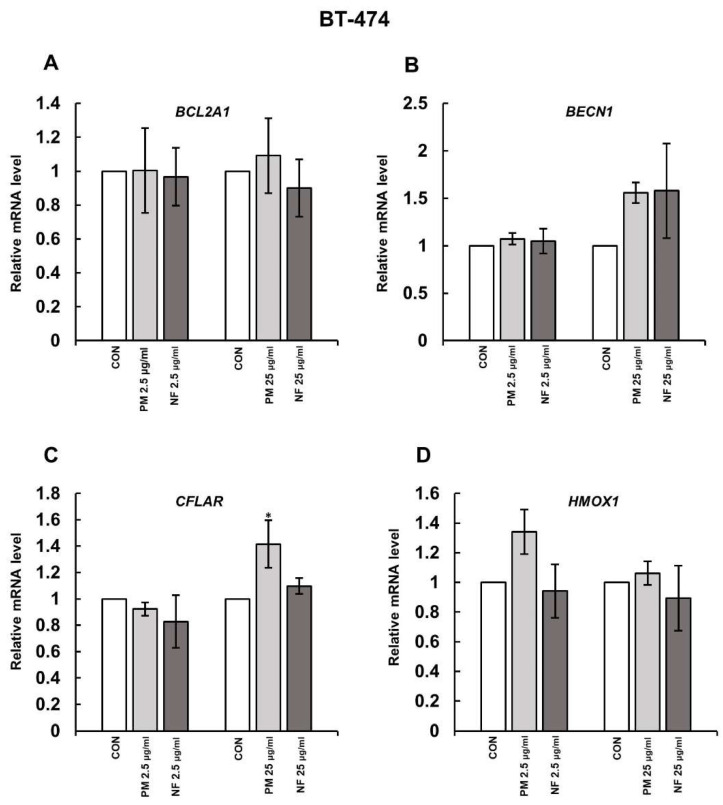
Dermatan sulfate (DS) rapidly stimulated the expression of *CFLAR* but not of other genes involved in cell survival in the luminal breast cancer cell line BT-474. The mRNA levels of *BCL2A1* (**A**), *BECN1* (**B**), *CFLAR* (**C**), and *HMOX1* (**D**) were quantified using RT-qPCR in the cell cultures that were treated for three hours with the tested DS variants (PM—DS from porcine intestinal mucosa; NF—DS from normal human fascia) at the indicated concentrations. The mRNA level for an individual gene was normalized to the endogenous human TATA-binding protein gene. The results are presented as relative mRNA (the mean ± SD) from four independent experiments. *—statistically significant differences (*p* < 0.05) versus the control.

**Figure 2 cells-14-01581-f002:**
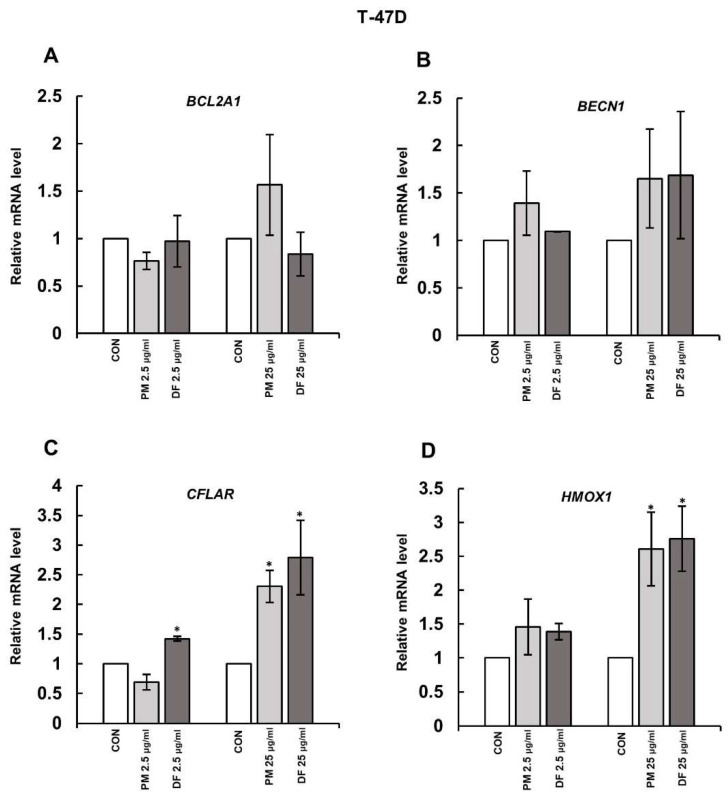
DS rapidly upregulated *CFLAR* and *HMOX1*, but not *BCL2A1* and *BECN1*, in the luminal breast cancer cell line T-47D. The mRNA levels of *BCL2A1* (**A**), *BECN1* (**B**), *CFLAR* (**C**), and *HMOX1* (**D**) were quantified using RT-qPCR in cell cultures that were exposed for three hours to the tested DS variants (PM—DS from porcine intestinal mucosa; DF—DS from fibrosis-affected human fascia) at the indicated concentrations. The mRNA level for each gene was normalized to the endogenous human TATA-binding protein gene. The results show relative mRNA levels for the tested genes and are presented as the mean ± SD of four independent experiments. *—statistically significant differences (*p* < 0.05) versus the control.

**Figure 3 cells-14-01581-f003:**
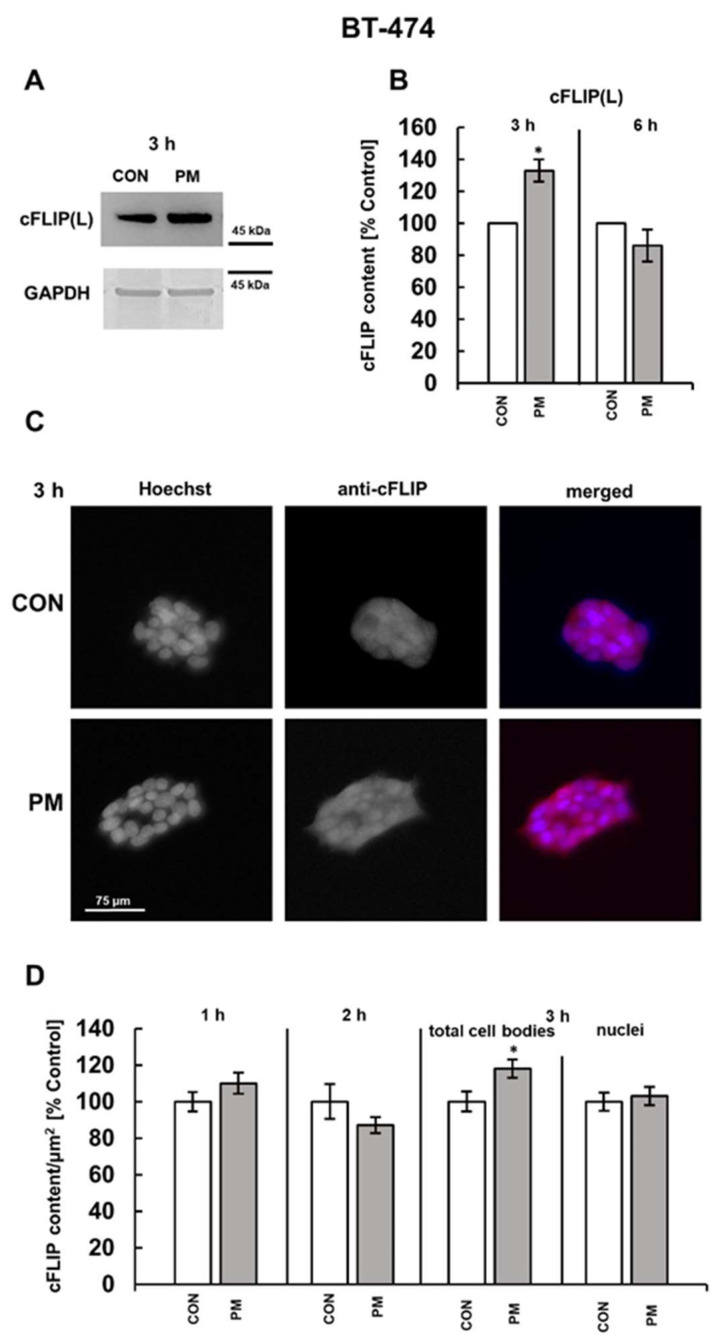
The PM variant rapidly increased the level of the long cFLIP splicing variant (cFLIP(L)) in BT-474 breast cancer cells. Cells were exposed for the indicated time periods to 25 µg/mL of the PM variant and then subjected to Western blotting after lysis in RIPA buffer (**A**,**B**) or to immunofluorescence (**C**,**D**). (**A**) Lysate aliquots (30 µg of proteins) were resolved by SDS-PAGE. cFLIP was immunodetected using the chemiluminescent ECL substrate, whereas GAPDH was detected immunologically using the colorimetric TMB substrate after blot stripping. Original blots are shown in Appendix A. (**B**) Quantification of the PM-triggered effect on cFLIP(L) expression in BT-474 cells after normalization of the protein level to the GAPDH content. (**C**) Representative fluorescence images showing the content and subcellular distribution (Appendix A) of cFLIP in the BT-474 cells that were treated with PM for the indicated time. cFLIP (red fluorescence) was detected using polyclonal antibodies at a 1:1000 dilution; nuclei (blue fluorescence) were stained by Hoechst. (**D**) Kinetics of the PM-induced effect on cFLIP expression in BT-474 cells, estimated by immunofluorescence. Cells were incubated with PM for the indicated time periods. cFLIP levels were quantified as the mean fluorescence per µm^2^ area of total cell bodies or nuclei in six non-overlapping fields from each experiment. Results in (**B**,**D**) are present as the mean ± SD of three independent blots or as the mean ± SEM of three independent experiments, respectively. *—differences statistically significant (*p* < 0.05) versus the control.

**Figure 4 cells-14-01581-f004:**
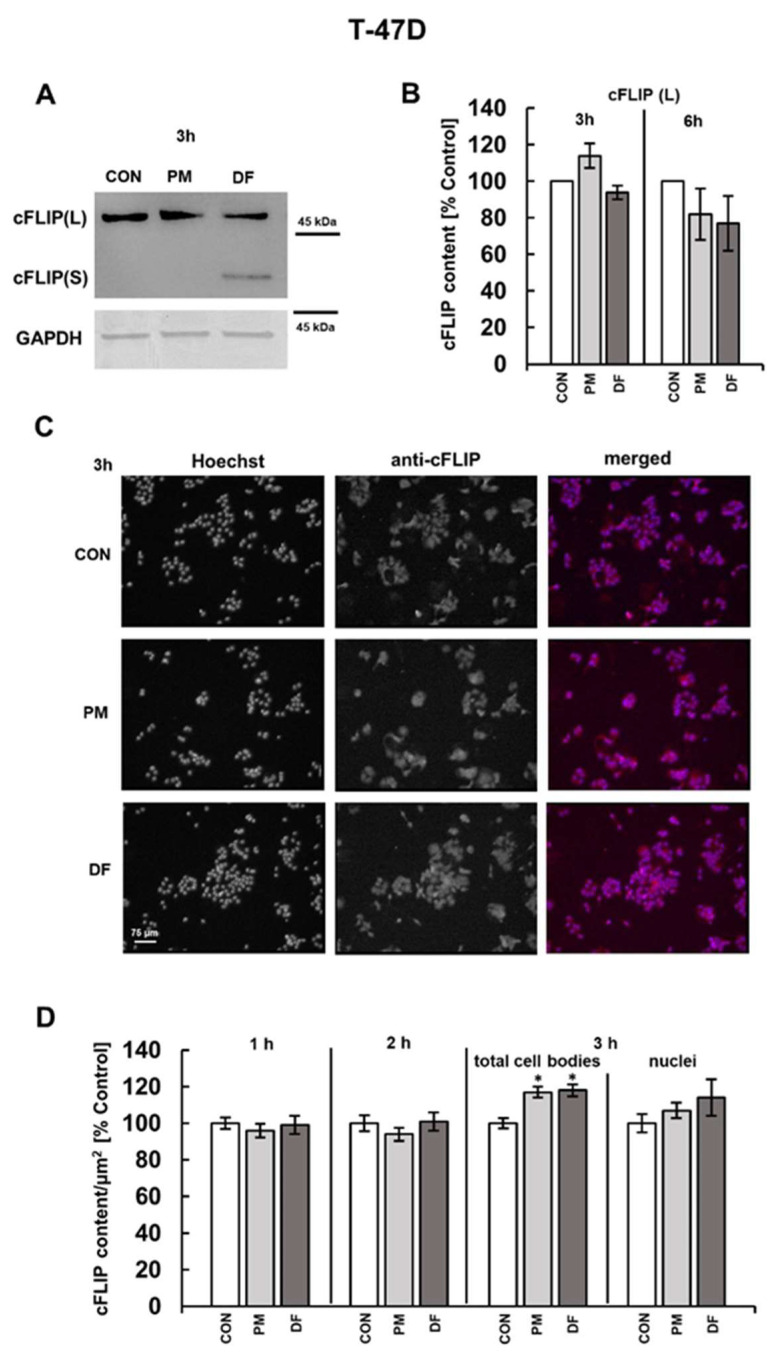
The PM and DF variants rapidly and selectively affected levels of the long or short splicing isoform of cFLIP, respectively, in T-47D breast cancer cells. Cells were incubated for the indicated time periods with 25 µg/mL of the PM or DF variant, and subsequently subjected to Western blotting after lysis in RIPA buffer (**A**,**B**) or to immunofluorescence (**C**,**D**). (**A**) Lysate aliquots (30 µg of proteins) were subjected to SDS-PAGE. The cFLIP splicing variants, i.e., long (cFLIP(L)) or short (cFLIP(S)), were detected immunologically using the chemiluminescent ECL substrate, whereas GAPDH was immunodetected using the colorimetric TMB substrate after blot stripping. Original blots are shown in Appendix A. (**B**) Quantification of the PM- or DF-induced effect on cFLIP(L) after normalization of the protein level to GAPDH. (**C**) Representative immunofluorescence images showing the content and subcellular distribution (Appendix A) of cFLIP (red fluorescence) in T-47D cells incubated with an individual DS variant for the indicated time periods. Nuclei were stained with Hoechst (blue fluorescence). (**D**) Kinetics of the PM- or DF-induced effect on cFLIP expression in T-47D cells, assessed by immunofluorescence. Cells were incubated with the DS variants for the indicated time periods. cFLIP levels was estimated as the mean fluorescence per µm^2^ area of whole cell bodies or nuclei for all of the visible cells in six non-overlapping fields from each experiment. Results in (**B**,**D**) are present as the mean ± SD of three independent blots or as the mean ± SEM of three independent experiments, respectively. *—differences statistically significant (*p* < 0.05) versus the control.

**Figure 5 cells-14-01581-f005:**
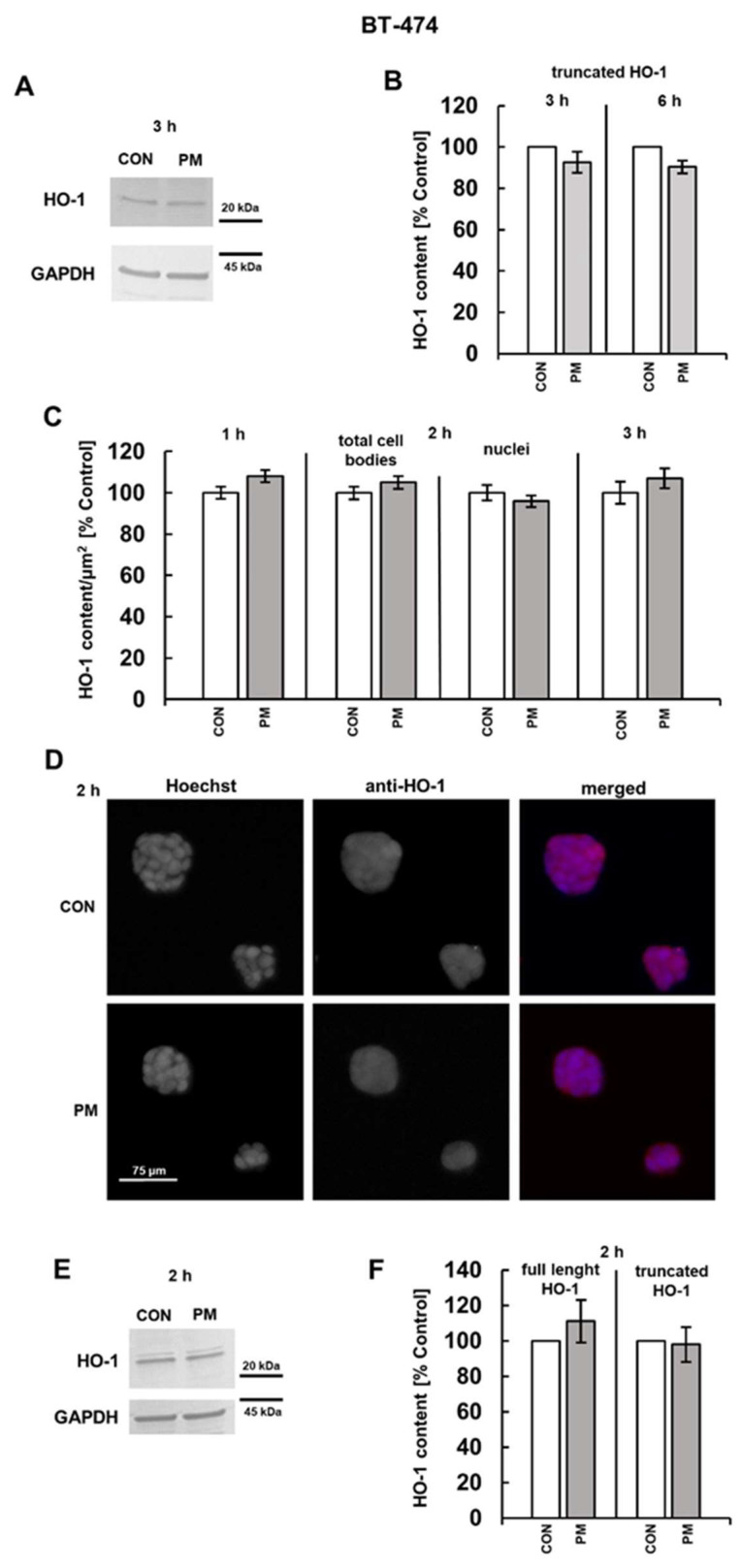
The PM variant did not affect either the level or subcellular distribution of heme oxygenase (HO)-1 in BT-474 breast cancer cells. Cells were exposed for the indicated time periods to 25 µg/mL of PM and then analyzed by Western blotting after lysis in RIPA buffer (**A**,**B**,**E**,**F**) or by immunofluorescence (**C**,**D**). (**A**,**E**) Lysate aliquots, containing 15 µg (**A**) or 30 µg (**E**) of proteins, were subjected to SDS-PAGE. HO-1 and GAPDH were immunodetected using the colorimetric TMB substrate. Original blots are shown in Appendix A. (**B**,**F**) Quantitative analysis of the immunoblots, showing the normalized HO-1 content in BT-474 cells that were treated with PM for the indicated time periods. (**C**) Dynamics of the PM-dependent effect on the HO-1 levels in BT-474 cells, assessed by immunofluorescence. The HO-1 content was estimated as the mean fluorescence per µm^2^ area of whole cell bodies or nuclei for all of the visible cells in six non-overlapping fields from each experiment. Results in (**B**,**F**) and (**C**) represent the mean ± SD of three independent blots or the mean ± SEM of three independent experiments, respectively. (**D**) Representative immunofluorescence images showing content and subcellular localization (Appendix A) of HO-1 (red fluorescence) in BT-474 cells that were exposed to PM for the indicated time periods. Nuclei were stained with Hoechst (blue fluorescence).

**Figure 6 cells-14-01581-f006:**
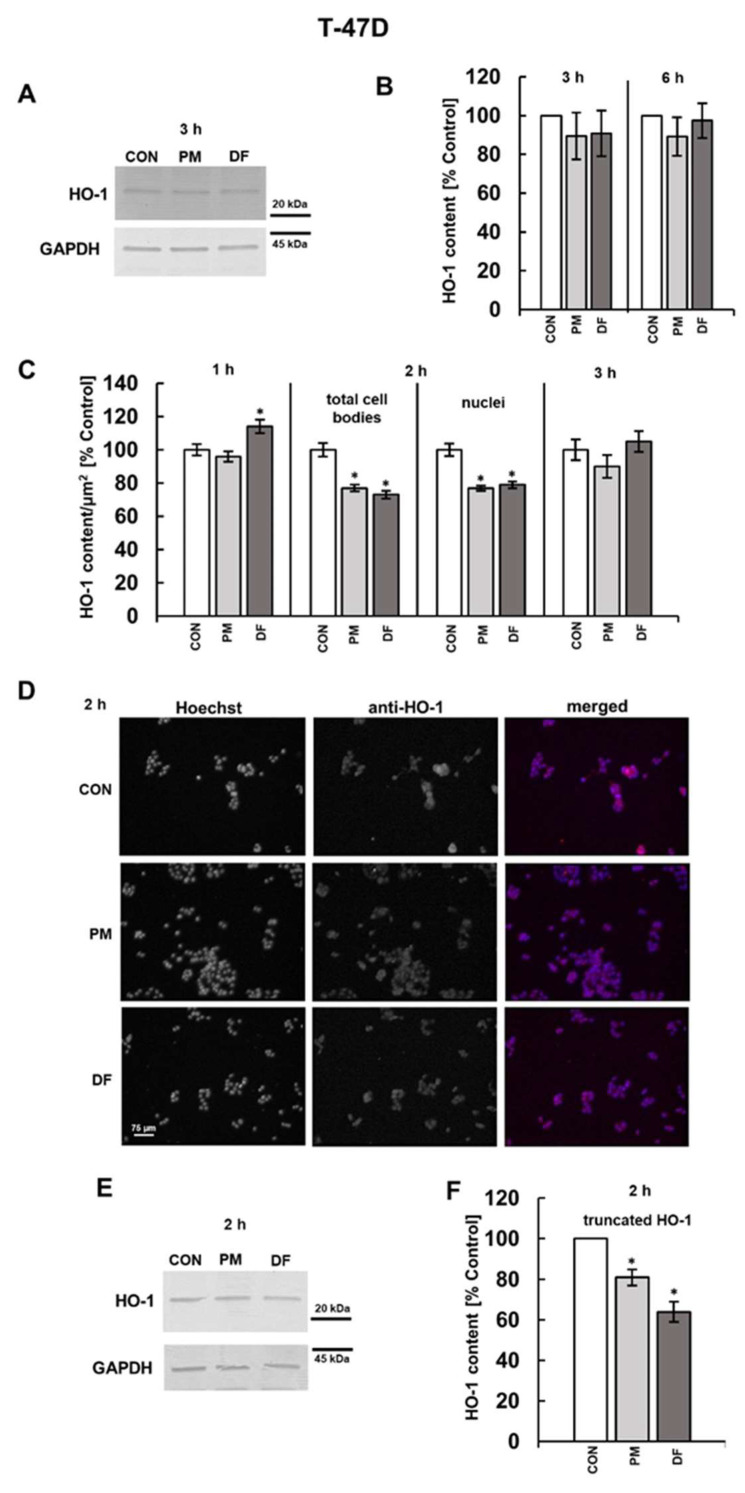
Both PM and DF modified the level and subcellular distribution of HO-1 in T-47D breast cancer cells. Cells were exposed to the PM or DF variant at 25 µg/mL for the indicated time periods and subsequently analyzed by Western blotting after lysis in RIPA buffer (**A**,**B**,**E**,**F**) or by immunofluorescence (**C**,**D**). (**A**,**E**) Lysate aliquots (15 µg of proteins) were separated by SDS-PAGE, followed by immunodetection using the TMB substrate. Original blots are shown in Appendix A. (**B**,**F**) Quantitative analysis of the immunoblots, showing normalized HO-1 content in the exposed T-47D cells. (**C**) Kinetics of the PM- or DF-triggered impact on HO-1 levels in T-47D cells, assessed by immunofluorescence. The HO-1 content was measured as the mean fluorescence per µm^2^ area of whole cell bodies or nuclei for all visible cells in six non-overlapping fields from each experiment. Results in (**B**,**F**) and (**C**) represent the mean ± SD of three independent blots or the mean ± SEM of three independent experiments, respectively. *—differences statistically significant (*p* < 0.05) versus the control. (**D**) Representative fluorescence images illustrating the influence of the PM or DF variant on the HO-1 content and subcellular localization of this protein (Appendix A) in T-47D cells after two hours of exposure. Cells were stained with anti-HO-1 antibody (dilution 1:200, red fluorescence) and Hoechst (nuclei, blue fluorescence).

**Figure 7 cells-14-01581-f007:**
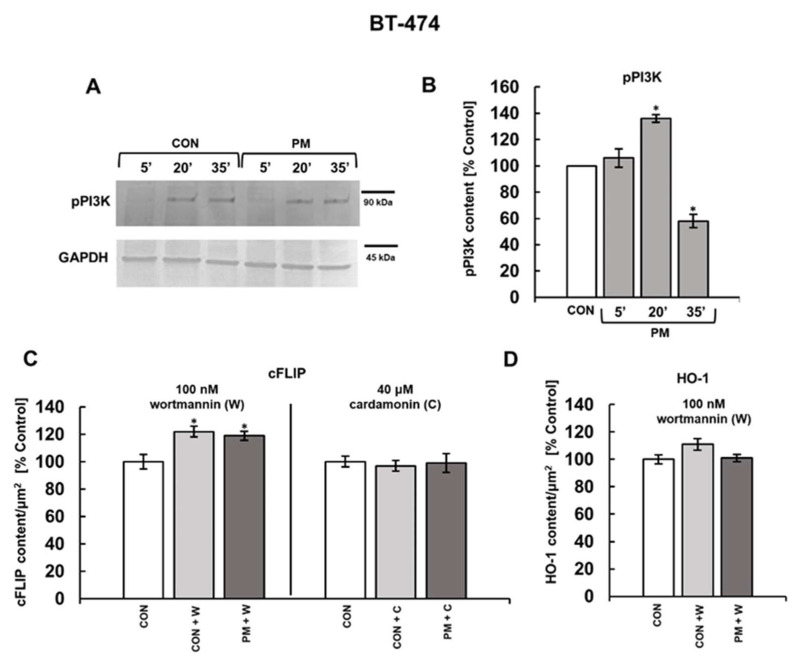
The PM variant regulated the cFLIP levels but not the HO-1 content in BT-474 cells via PI3K and NFκB signaling. (**A**,**B**) PM modified the activation of PI3K in BT-474 cells. Cells were exposed to this variant for the indicated time periods, lysed in RIPA buffer and subjected to SDS-PAGE (**A**). The obtained blots were probed with polyclonal anti-phospho-PI3K antibodies at a dilution of 1:1000. Original blots are shown in Appendix A. (**B**) Quantification of dynamics of the PM-induced activation of PI3K in BT-474 cells. Phospho-PI3K levels were normalized to GAPDH. (**C**,**D**) Pharmacological inhibition of both PI3K and NFκB signaling affected the PM-dependent increase in cFLIP level in BT-474 cells. Cells were exposed to PM in a combination with wortmannin (W, an inhibitor of PI3K activity) or cardamonin (C, an inhibitor of NFκB pathway) at the indicated concentrations for three (**C**) or two (**D**) hours. The influence of pharmacological inhibition on the PM-exerted effect on the cFLIP (**C**) or HO-1 (**D**) levels was assessed by immunofluorescence. Representative images are shown in Appendix A. Results in (**B**) and (**C**,**D**) represent the mean ± SD of two independent blots or the mean ±SEM of at least six images in each of the two independent experiments, respectively. *—differences statistically significant (*p* < 0.05) versus the control.

**Figure 8 cells-14-01581-f008:**
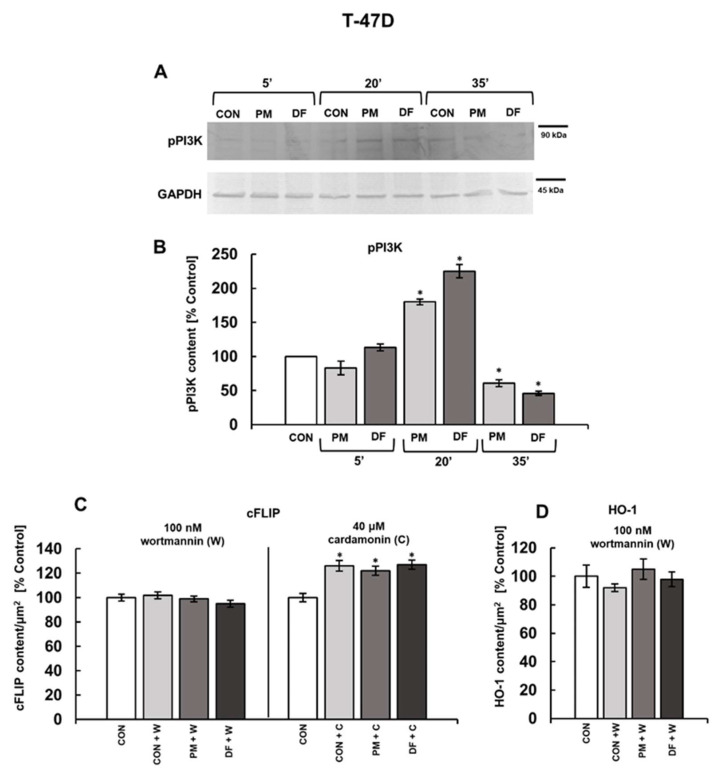
Both the PM- and DF-triggered effects on the cFLIP and HO-1 levels in T-47D breast cancer cells were mediated through PI3K and/or NFκB signaling. (**A**,**B**) Both DS variants significantly modulated PI3K activation in the breast cancer cells. (**A**) Cells were grown for the indicated time periods in the presence of PM or DF at a concentration of 25 µg/mL, and then they were lysed in RIPA buffer and subjected to SDS-PAGE, followed by immunodetection using first anti-phospho-PI3K antibodies, and, after stripping, anti-GAPDH antibodies. Original blots are shown in Appendix A. (**B**) Quantification of dynamics of the DS variant-induced activation of PI3K in T-47D cells. Levels of phospho-PI3K were normalized to GAPDH content. (**C**,**D**) Effects of pharmacological inhibition of PI3K or NFκB pathways on the DS variant-induced changes in the expression of cFLIP or HO-1, assessed by immunofluorescence. Cells were treated with a combination of an individual DS variant and wortmannin or cardamonin at the indicated concentrations for three (**C**) or two (**D**) hours. Representative images are shown in Appendix A. Results in (**B**,**C**) and (**D**) represent the mean ± SD of two independent blots or the mean ± SEM of at least six images in each of two independent experiments, respectively. *—differences statistically significant (*p* < 0.05) versus the control.

**Table 1 cells-14-01581-t001:** KiCqStart primer sequences specific to targeted genes.

Gene	Primers
*BCL2A1*	Forward 5′-CAAGAAACTTCTACGACAGC-3′ Reverse 5′-AAGCCATTTTCCTCTTCTTG-3′
*CFLAR*	Forward 5′-AGGGACAAGTTACAGGAATG-3′ Reverse 5′-GAGCCTGAAGTTATTTGAAGG-3′
*BECN1*	Forward 5′-ATGAGATTAATGCTGCTTGG-3′ Reverse 5′-AGAGACTCCAGATATGAATGG-3′
*HMOX1*	Forward 5′-CAACAAAGTGCAAGATTCTG-3′ Reverse 5′-TGCATTCACATGGCATAAAG-3′

## Data Availability

Data will be shared upon request (please contact with mkozma@sum.edu.pl).

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
