# Peer review of "Structural Variants of Dermatan Sulfate Can Affect the Expression of Proteins Involved in Breast Cancer Cell Survival"

_cells, 2025, doi:10.3390/cells14201581_

Round 1
Reviewer 1 Report
Comments and Suggestions for Authors
It is an interesting study that offers new insights in the effects of DS. It has been well designed and written with clear messages. However, this reviewer feels that both the intro and the discussion parts could be substantially improved in quality and structure of the manuscript. This is because of there are references that should be added to support hypothesis and conclusions. These are background references dealing with the several years ago discoveries on the sequence analysis of DS by various analytical assays and differential emzymatic cleavages and in addition recent critical reviews dealing with the ecm structure and function, and the the biological roles of GAGs. This could be discussed and improve the quality of the manuscript.
Reviewer 2 Report
Comments and Suggestions for Authors
Wisowski and colleagues extend their studies on the molecular mechanisms involved in the DS-dependent induction of necroptosis in breast cancer cells by assessing the impact of different DS structural variants on several proteins implicated in the regulation of breast cancer cell survival. Although the study is well-organized and interesting, the relevance of the findings is questionable since DS cannot exist in free form but as a side chain of a proteoglycan (for example decorin). Therefore, the functional properties of natural occurring DS are critically affected and mediated by the accompanying core protein. Therefore, it would be extremely interesting and biological relevant if these structural variants of DS were studied as part of a proteoglycan instead of being free GAGs.
Major:
- The authors used three structural variants of DS that were previously characterized in terms of their structural features in Wisowski et al. (2022). These isoforms included a commercial preparation of DS from porcine intestinal mucosa (PM) and two variants of human origin: DS from human fibrosis-affected palmar fascia (DF) and from normal human fascia lata (NF). The authors should state in detail the extend of the sulfation pattern (i.e. unsulfated, 4-S, 6-S, 4,6-S) and glucuronosyl epimerization (GlcA to IdoA ratio) levels of each DS variant to better understand the DS structure-dependent effects in the context presented in this study.
- In Results section, the authors state that “…the tested DS variants were used at two concentrations, i.e. 25 μg/mL and 2,5 μg/mL. The former concentration was known to effectively trigger necroptosis of breast cancer cells [Refs], whereas the second concentration was used to compare the biological effectiveness of DS”. Why two concentrations of DS variants were used and what do the mean by “biological effectiveness”? Isnt’t necroptosis considered an outcome of such a biological effectiveness?
- A better quality and higher magnification of IF images are required especially for the monitoring of subcellular distribution of proteins.
- Fig. 1: None of the tested DS variants affected proteins relevant to cell viability in BT-474 cells apart from the PM that stimulated CFLAR mRNA expression at high concentration.
Fig. 2: Both DS variants (at high concentrations) upregulated CFLAR and HMOX1 mRNA expressions in T-47D cells.
Figs. 3/4: PM increased CFLIP protein expression in BT-474 cells but not T-47D (as authors state) at high concentrations. Moreover, DF had no effect on CFLIP(L) but strongly upregulated CFLIP(S) in T-47D cells.
The authors conclude that regardless of the DS variant or cancer cell line, this glycan (DS) only triggered a sudden and transient upregulation of CFLIP (3h post-treatment). However, this conclusion is not in agreement with the results since there are differences in the effects of the different DS variants (PM vs DF) as well as cancer cell line (BT-474 vs T-47D cells) and not only at CFLIP levels. This conclusion should be re-considered by the authors. They should also provide explanations for the observed differences between DS variants and between cell lines.
- HO-1: PM and DF (3h exposure) upregulated the HMOX1 mRNA expression in T-47D cells (Fig. 2D). However, none of the DS variants significantly influenced the protein levels of HO-1 in T-47D cells. When DS variant DF was tested for a shorter time (1 hr) upregulated HO-1 protein (Fig. 6C), while 2h of exposure to both PM and DF resulted in the significant reduction of HO-1 protein levels in T-47D cells (Fig. 6C, D).
How do the authors explain these discrepancies in mRNA and protein levels of HO-1?
Minor:
- The introduction is too long and should be shortened.
- In Introduction section:
- i) “galactosaminglycans” should be corrected to “galactosaminoglycans”
- ii) “…DS-bearing glycoprotein, known as decorin…” should be corrected to “…DS-bearing proteoglycan, known as decorin…”
Reviewer 3 Report
Comments and Suggestions for Authors
The review of the manuscript „Structural variants of dermatan sulfate can affect the expression of proteins involved in breast cancer cell survival”
The manuscript explores the molecular mechanisms by which dermatan sulfate (DS) variants affect regulators of apoptosis, autophagy, and oxidative stress in luminal breast cancer cell lines. The topic is relevant and novel, and the experiments are generally well-designed. However, there are some points to be improved.
- The abstract is informative but too dense; it would benefit from clearer highlighting of novelty and the main mechanistic findings (e.g., isoform-specific effects on cFLIP).
- The introduction provides background but could be streamlined. Consider shortening the general glycosaminoglycan section and expanding the rationale for why cFLIP and HO-1 are particularly relevant in luminal breast cancer.
- Primer sequences in Table 1 appear to be duplicated and may contain errors (identical sequences for all genes). Please check and correct.
- Figures 1–6: The legends are long but sometimes miss key clarifications (e.g., which cFLIP isoforms are shown in Western blots). Consider simplifying the presentation and emphasizing the structure-dependent effects.
- Some findings (e.g., HO-1 upregulation at 1 h, downregulation at 2 h, then recovery) are intriguing but need more quantitative support. Were time-course experiments statistically robust across replicates?
- The discussion is very detailed but occasionally speculative. Please better distinguish between data-supported conclusions and hypotheses.
Round 2
Reviewer 2 Report
Comments and Suggestions for Authors
The authors have adequately responded to the reviewer's concerns. In my opinion, the ms has been improved.
Minor:
- Table 1: "Foreward" should be corrected to "Forward"
- Figs 3,4,5,6: IF images are note coloured as indicated in relevant legends (for example, in Fig. 3C, cFLIP (red fluorescence) and nuclei (blue fluorescence) stained by Hoechst).
Author Response
We would like to thank the Referee for all valuable comments provided in both Round 1 and 2. There is no doubt that these remarks enabled us to significantly improve the quality of our manuscript. Below we present detailed answers to the latest comments (Round 2)
Minor comments
Ad.1
The indicated error in Table 1 has been corrected
Ad. 2
In the indicated figures, some panels showing single-type fluorescence (i.e. cFLIP immunofluorescence or nuclear staining) are presented in black and white to better highlight quantitative differences between treatments. However, panels with merged fluorescence retained their original colors, whose sources need to be explained (blue fluorescence – nuclei, red fluorescence – cFLIP). With the Reviewer’s permission, we would like to retain this information in the figure legends.